# Microcosm Study on the Potential of Aquatic Macrophytes for Phytoremediation of Phosphorus-Induced Eutrophication

**Sarah Dean [1], Muhammad Shahbaz Akhtar [1,\*], Allah Ditta [2,3], Mohammad Valipour [4,\*] and Sohaib Aslam [1]**

1 Department of Environmental Sciences, Forman Christian College University, Lahore 54600, Pakistan
2 Department of Environmental Sciences, Shaheed Benazir Bhutto University, Sheringal, Dir (U) 18000, Pakistan
3 School of Biological Sciences, University of Western Australia, Perth, WA 6009, Australia
4 Department of Engineering and Engineering Technology, Metropolitan State University of Denver, Denver, CO 80217, USA
\* Correspondence: shahbazakhtar@fccollege.edu.pk (M.S.A.); mvalipou@msudenver.edu (M.V.)

**Abstract:** Phosphorous (P) is one of the primary nutrients to cause the eutrophication of water bodies. This process leads to algal blooms and anoxic conditions which have consequences in the form of mortality of aquatic animals, and impaired water quality. Aquatic macrophytes could be the promising candidates that can filter P from water contaminated with high levels of nutrients. In the present microcosm research, two types of floating macrophytes, i.e., salvinia floater (*Salvinia natans*) and water lettuce (*Pistia stratiotes*) were deployed to compare their P-removal rates and efficiency under different incubation times (72, 168, and 264 h intervals). Plants were exposed to different treatments, i.e., (1) P-fed plants, (2) P-starved plants, (3) control treatments, and (4) synthetic wastewater treatment. Both plant species showed substantial P-removal efficiency from P-eutrophicated solutions and removed P-amounts were significantly correlated ($R^2 \cong 1$ at $p \leq 0.05$) with P-accumulated in plant biomass. Plants in the P-starved state showed significantly higher P-removal rates and removal efficiency compared to plants without P-starvation. When *Salvinia natans* was exposed to 10 mg L$^{-1}$ of P for 264 h of incubation, 21 g of fresh biomass was recorded during the P-starved phase, more than *P. stratiotes* (14 g) under similar conditions. The *P. stratiotes* removed 86.04% of P from 5 mg L$^{-1}$ P solution, 53.76% from 10 mg L$^{-1}$ P solution and 66.84% from SWW in the P-starved phase whereas, removal efficiency without the P-starvation phase was 33.03% from 5 mg L$^{-1}$ P solution, 39.66% from 10 mg L$^{-1}$ P solution, and 31.64% from SWW after 264 h interval. Compared to *S. natans*, *P. stratiotes* removed 86.0% P from a 5 mg L$^{-1}$ P solution, whereas *S. natans* removed 56.6% when exposed to the same P solution (5 mg L$^{-1}$ P solution). Bioconcentration factor (BCF) values were higher in *Salvinia natans* 10.5 (0.5 mg L$^{-1}$ P solutions) and 1.5 (5 mg L$^{-1}$ P solutions) compared to 9.9 and 1.3 of *Pistia stratiotes* under P-starved conditions. The present work highlighted that these aquatic plants can be a potential green sustainable solution for purifying water with excessive nutrients (N and P), especially waters of wetlands, lagoons, and ponds.

**Keywords:** macrophytes; phytoremediation; water lettuce; salvinia floater; P-eutrophication; P-starvation

## 1. Introduction

Phosphorous (P) is the 11th most abundant element present in the earth's crust and enters the soil and water by weathering of rocks or by anthropogenic activities like mining [1]. It is not found as a free element in the biosphere and generally occurs as "phosphate", i.e., compounds that contain phosphate ion, $PO_4^{3-}$ in minerals. Phosphate is also found in living organisms as part of their DNA (deoxyribonucleic acid), RNA (ribonucleic acid) and ATP (adenosine triphosphate), cell membrane (phospholipids), and it can be recovered from their bones and urine [2]. The average consumption of P per person is about 1.2 g day$^{-1}$, which accounts for 8300 tons of P for the global population per year [3]. Phosphorous demand is rising day by day due to its use in commercial products like toothpaste,



pesticides, fertilizers, match sticks, and military bombs like hand grenades. Phosphorous is used in detergents heavily in the form of builders and is commonly used as dry detergents (sodium tripolyphosphate) or liquid detergents (potassium/sodium phosphates). These P-rich detergents have a great potential to pollute surrounding water bodies and most of these detergents are persistent because of the presence of alkyl benzene sulfonate. Phosphorus is a non-renewable resource that is expected to deplete in 100 to 300 years if not used efficiently [3]. As fossil fuels can be replaced with renewable or biodegradable fuels, phosphorous cannot be replaced with anything. It can only be used vigilantly or with recycling.

Agriculture is one of the main reasons for phosphorous depletion [4]. Environmentalists are finding different ways to extract phosphorous from the already excessive P-contaminated wetlands, municipal water, animal waste, and agricultural runoffs. Many environmentalists suggested that the use of aquatic plants might be the best way to recycle this mineral eco-friendly in the form of fertilizers and to stop the hazardous algal blooms from excess agricultural runoff [5].

Fertilizers rich in phosphorous (P) are frequently used for high-yield crops in agricultural ecosystems to overcome the problem of low P availability. The long-term use of these fertilizers causes a significant buildup of NPK in soil and when soils become saturated with nutrients mainly with N and P; soils are subjected to losses by surface runoff or leaching. The runoff carrying the nutrients may end up in ponds, lakes, lagoons, reservoirs, bays, rivers, or wetlands. Eutrophication decreases water quality and when algae die, it is decomposed by microorganisms present in water and is converted into inorganic form; this conversion depletes dissolved oxygen in the water and may lead to "fish kills" by creating an anoxic zone in water body [6–8]. The detergents and manure contain higher concentrations of phosphorous and nitrogen whereas internal factors are associated with sediments present in the water bodies which release phosphorus because of recycling processes [8,9].

Among the plethora of remediation techniques, phytoremediation is a solar-driven green technique. This technology uses diverse plant species as well as macrophytes along with associated rhizospheric microorganisms to degrade or sequester the inorganic or organic pollutants through their enzymes and metabolites/substrates [10–12]. Phytoremediation is encompassing many technologies, such as phytodegradation, phytostimulation, phytoaccumulation, phytostabilization, and phytovolatilization, which may occur simultaneously [13]. In this regard, aquatic macrophytes can be deployed effectively for the phytoremediation of pollutants and nutrients present in water bodies as these could serve as phytoindicators due to their response towards light, nutrients, contaminants, toxins, metals, and salts [14]. These plants have the potential to produce higher biomass under contaminated environments which results in their higher efficiency in a very short time [15]. Moreover, these plants have the potential to remediate different contaminants via phytodegradation, phytoaccumulation, phytostabilization, and phytovolatilization [16,17].

Among macrophytes, *Pistia stratiotes* commonly known as water lettuce and sometimes water cabbage is originally native to Texas and Florida but now is found in the majority of warmer regions of the world. It reproduces through stolen (the propagation strategy, where the new individual is formed through the mother plant) and its leaves are organized like rosettes. The foliage of this plant can grow approximately six inches tall but its roots can be 20 inches deep [18]. *Pistia stratiotes* might be an efficient macrophyte for removing and accumulating P because it can double biomass within a month. In an experiment on Kabar wetlands in India containing dissolved inorganic phosphorus (DIP) of approximately 0.67 mg $L^{-1}$ maximum in October and 0.05 mg $L^{-1}$ minimum in June, seven basins were experimentally constructed to test *Pistia stratiote* ability [19]. The tested plant showed its peak growth in September and accumulated about 45 mg $L^{-1}$ of phosphate in 35 days. In another study, the *Pistia stratiotes* were tested on sewage water for 10 days. It also reduced total phosphorus (TP) and $PO_4^{3-}$ about 14–31% and removed 25–34 kg P $ha^{-1}$ annually [7]. *Pistia stratiotes* has higher P removal efficiency due to its long roots (capable of

high rhizofiltration) that are approximately 49 cm in length on an average basis. In a case study performed at the National Institute of Oceanography and fisheries in Alexandria, Egypt, twelve plastic tubs were taken, and 10 cm of soil was filled in containers with 40, 70, and 90 L of wastewater. The 10, 13, and 17 plants of water lettuce were used in an experiment to depict 50, 70, and 90% surface coverage. The highest removal rate of P was on the 5th day but for 90% plant coverage of about 87%, while for 70% plant coverage the highest removal day was the 4th day (84.5%) and for 50% plant coverage the highest removal day was 3rd day (80.3%). The best depth for the highest removal rate was 25 cm [20].

*Salvinia natans* commonly known as salvinia floater flourish in the shade, which produces a thick mat [21]. It is usually present in slow-moving or stagnant waters. It is associated with tropical or temperate zones. It is a floating fern species, and it does not have any roots. Its leaves are divided into two parts- one that is floating and the other is a submerged hairy leaf, which functions like a root. It produces leaves in pairs. *Salvinia natans* has also been reported to remove high TP, about 67.3–90.6% in one study as compared to other species of aquatic plants (*Ipomoea aquatica*, *Eichhornia crassipes*, *Hydrocotyle vulgaris*, *Eleocharis plantagineiformis*, *Colocasia tonoimo*, *Dysophylla sampsonii*, *Typha orientalis*, and *Rotala indica*) in synthetic domestic wastewater [22]. Salvinia floater could be a potential candidate for phytoremediation of the unhealthy environment in aquatic ecosystems containing higher amounts of nitrogen and phosphorous [14]. Nevertheless, both *Salvinia natans* and *Pistia stratiotes* were not tested to estimate their phytoremediation potential when plants in a P-starved state were exposed to P-eutrophicated aquatic media. The present study was conducted to compare the effectiveness of two aquatic macrophytes species, i.e., salvinia floater (*Salvinia natans*) and water lettuce (*Pistia stratiotes*) for phosphorous removal through serial microcosm experiments. The efficiency of P filtration of species along with the potential of P-starved and P-fed plants to bioaccumulate P in their biomass was evaluated. The phosphorous uptake rate in P-starved plants as well as in P-fed plants was also estimated.

## 2. Materials and Methods

### 2.1. Experimental Protocol and Plant Culture Environment

Cultural Conditions and Experimental Set-Up

Macrophytes were cultured in a container/plastic tub of 50 L capacity in the Botanical Garden of Forman Christian College University, Lahore (31°31′19.988″ N, 74°20′4.3908″ E). The container was filled with 10 kg soil and 20 L tap water to get reasonable biomass for 8 months with continuous water and soil change. The soil used for the growth of macrophytes was taken from Forman Christian College's Botanical Garden. A wooden structure was made and a net was laid on top of the structure to protect the plants from outside contamination like fallen leaves, bird excreta, etc. In harsh weather conditions like fog and rain, a plastic sheet was laid on top of the net to protect plants from harsh conditions. The plants were cultured in December 2020. Plants doubled in size by May 2021. Nevertheless, aquatic plants are fast-growing, but their growth was slightly hindered due to cold weather in the winter season. However, once the weather was warm in March 2021, the plants started to grow faster. The experiment started in July 2021. The plants for the experiment were transferred to the laboratory once the desired biomass was achieved. The serial microcosm experiments started on 27 July 2021, and ended on 31 September 2021.

Experiments were conducted with initial concentrations of 0.5, 1, 5, and 10 ppm of P using a modified Hoagland solution with three replications of each treatment. In addition, three sets of plants were cultured in distilled water only without Hoagland and phosphorous and were labeled as DW for distilled water. The other set was cultured in P-free Hoagland solution and was labeled 0 which means the absence of phosphorous and the last set of plants was cultured in synthetic wastewater (SWW). Synthetic wastewater was used for this study to investigate the response of tested plants exposed to P in presence of other metals. The synthetic wastewater was prepared and used after considering the

average metal and pollutant concentration in domestic wastewater collected and tested from the surrounding drains and sewerage system of Lahore, Pakistan [23]. The composition of SWW is represented in Table 1. For comparison, the control group with Hoagland solution and no plant was placed next to the experiments. The composition of the modified Hoagland solution is represented in Table 2. One experimental set of the macrophytes was exposed to the P-fed directly after washing them with distilled water and the other set was re-cultured in distilled water for a week before their exposure to P-induced water. This was done to starve the plants to estimate whether plant uptake was increased or not under P-starvation. There were three replications of each set.

**Table 1.** Composition of synthetic wastewater (SWW) used in experiments.

| Salts | Salt Names | Quantity (g 5 L$^{-1}$) |
|---|---|---|
| $C_6H_{12}O_6$ | Glucose | 3.60 |
| $C_{13}H_{24}O_4$ | Peptone | 2.70 |
| $C_{85}H_{12}4N_{14}O_{16}S$ | Yeast | 0.36 |
| $(NH_4)_2SO_4$ | Ammonium Sulfate | 2.88 |
| $KH_2PO_4$ | Potassium dihydrogen phosphate | 0.50 |
| $MgSO_4 \cdot 7H_2O$ | Magnesium sulfate heptahydrate | 0.72 |
| $MnSO_4 \cdot 7H_2O$ | Manganese sulfate heptahydrate | 0.0648 |
| $FeCl_3 \cdot 6H_2O$ | Ferric chloride hexahydrate | 0.0036 |
| $NaHCO_3$ | Sodium Bicarbonate | 9.00 |
| $CaCl_2 \cdot 2H_2O$ | Calcium chloride dihydrate | 0.072 |

**Table 2.** Modified 1% Hoagland nutrient solution composition used in the present study.

| Salts | Salt Names | Quantity (mg L$^{-1}$) |
|---|---|---|
| $NH_4NO_3$ | Ammonium Nitrate | 2.00 |
| $CaCl_2 \cdot 2H_2O$ | Calcium chloride dihydrate | 2.06 |
| $KCl$ | Potassium Chloride | 2.00 |
| $MgSO_4 \cdot 7H_2O$ | Magnesium sulfate heptahydrate | 0.48 |
| $MnSO_4 \cdot 5H_2O$ | Manganese sulfate pentahydrate | 0.005 |
| $EDTA\text{-}Na\text{-}Fe \cdot H_2O$ | Ethylenediaminetetraacetic acid disodium | 0.062 |
| $ZnSO_4 \cdot 6H_2O$ | Zinc sulfate hexahydrate. | 0.007 |
| $H_2MoO_4 \cdot H_2O$ | Molybdic acid monohydrate | 0.005 |
| $CuSO_4 \cdot 5H_2O$ | Copper Sulfate pentahydrate | 0.006 |
| $H_3BO_3$ | Boric Acid | 0.005 |

Since the species were different, so they were weighed and sized manner uniformly before culturing these species in P-induced water. Weighing and pruning of macrophytes were done for phytofiltration studies. The macrophytes were pruned to cut down their weight to 10 g so that the uniform size and weight of plants should be exposed to different treatments. For study 1, plants were transferred to 1000 mL plastic containers containing 1000 mL of P-free with 1% modified Hoagland medium having a pH of 6.5. The plants were placed at room temperature (30 ± 2 °C) under the artificial lights, having a light intensity of 50 to 70 candela. The lights were adjusted with the timer to turn the lights off after 12 h, i.e., the lights turned on at 6:00 am and turned off at 6:00 pm. The lights were mounted on the wooden frames and the containers were placed below the frame for maximum light exposure. The plastic containers were covered with plastic cover with approximately 100 cover holes/cover for evapotranspiration purposes. The experimental setup is represented in Figure 1.

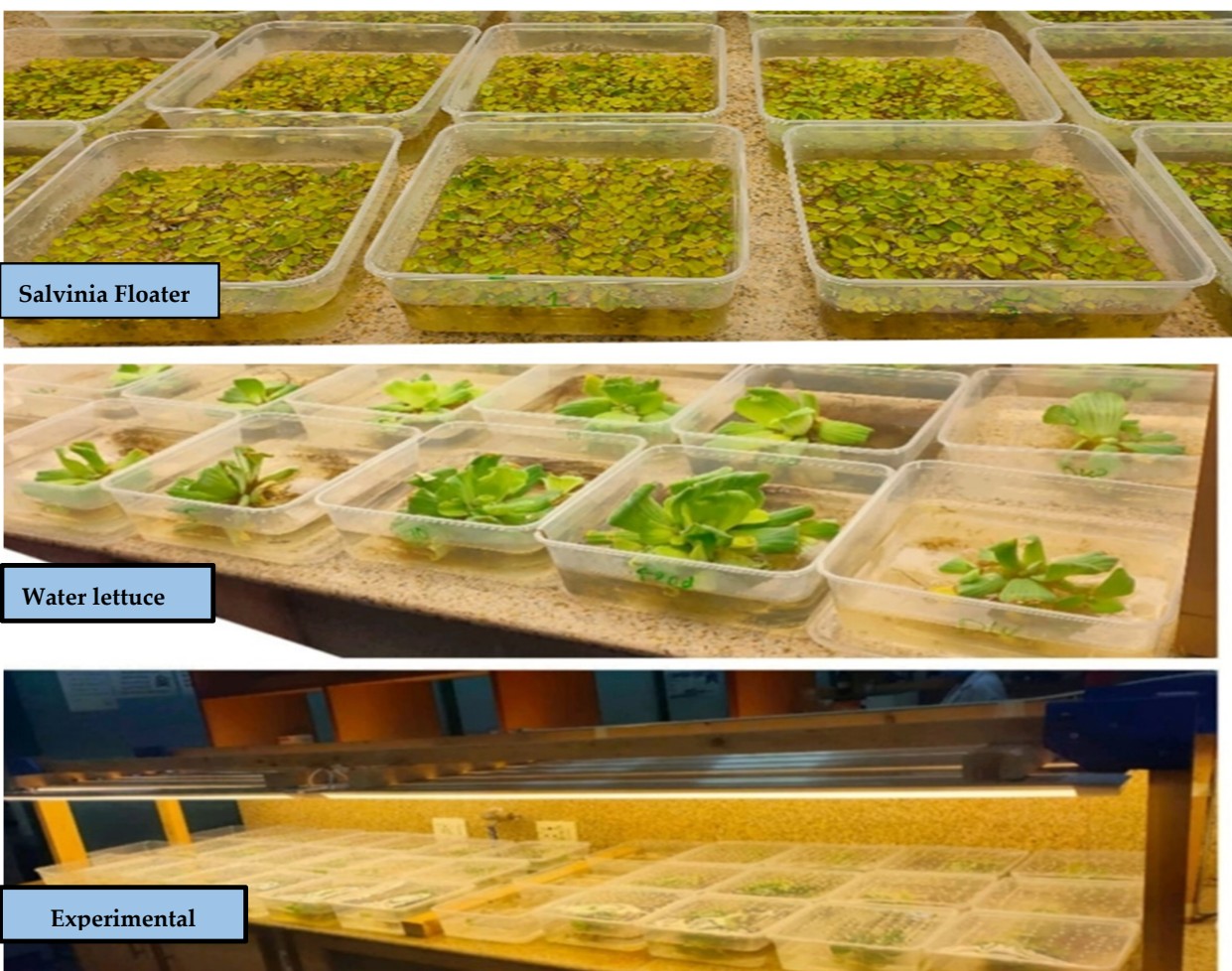

**Figure 1.** Macrophytes submerged in their respective treatments.

*2.2. Plant Biomass and Phosphorous Assay*

All plants were harvested at 72, 168, and 264 h intervals during the incubation period. The incubation was followed by the filtration of samples via a 150-micron sieve and their wet biomass was recorded on weighing balance. The phosphorous assay of water samples was done through the molybdenum blue method using the spectrophotometry technique. Spectrophotometry is a technique that uses orthophosphate to form molybdophosphoric acid. The excess molybdate is reduced in acidic conditions to give molybdenum blue color. The absorbance of molybdenum blue is then recorded using a spectrophotometer at 880 nm for phosphorous. The intensity of the blue color is directly proportional to the amount of P present in solutions [19]. The plants after their incubation periods were then put in brown craft paper bags and dried in an oven at 80 °C for 24 h. The dry biomass (g container$^{-1}$) was recorded. The dried plants were ground to form a fine powder with the help of a Mill and 0.1 g of powdered sample was transferred into porcelain crucibles, which were then placed into a Muffle furnace where the temperature was gradually increased to 550 °C for 12 h for ashing process. After 12 h, the ash was cooled down at room temperature. Afterward, the ash was mixed with 5 mL of 2N hydrochloric acid (HCl). The crucibles were then placed on the hotplate to evaporate the HCl at 80 °C temperature until the formation of pellets [1]. These pallets were dissolved again in 5 mL of 2N hydrochloric acid to get a clear solution. The solutions were then filtered via filter paper in a 100 mL flask and the flask was filled to the mark with demineralized water. The phosphorous concentrations in solutions were measured in "mg L$^{-1}$" and were analyzed through the molybdenum blue method using the spectrophotometry technique. The operational conditions of this spectrophotometric method are previously reported by Akhtar et al. [1].

### 2.3. Phosphorous Analysis and Working Solution

The phosphorous stock solution was formed to make phosphorous standards of 0.5, 1, 5, and 10. For this 5.741 g of $Na_2H_2PO_4 \cdot 12H_2O$ salt was mixed in 1000 mL of distilled water and named as $G_1$ stock solution. Then, 10 mL from the $G_1$ stock solution was taken into 100 mL in a different volumetric flask and named as $G_2$ sub-stock solution. Working solutions of 0.5, 1, 5, and 10 ppm were prepared from $G_2$ sub-stock solution. About 20–30 min after the addition of color developing reagent, absorbance was measured at 880 nm using a UV spectrophotometer. The combined reagent commonly known as the color-developing reagent is used for the determination of phosphorous in the water. The water can be from any source, that can be drinking, industrial, or any surface water. Removal of phosphorous was calculated through the following calculations.

### 2.3.1. Amount of P Removed

The amount of P removed was calculated from the equation given below:

$$\text{Amount of P removed} \left( \text{mg m}^{-2} \right) = \frac{(W_i \times P_i) - (W_f \times P_f)}{\text{Surface area}}$$

where $W_i$ = initial mass of water, $P_i$ = initial concentration of phosphorus, $W_f$ = mass of water at the end of the experiment, and $P_f$ = final concentration of phosphorus.

### 2.3.2. Removal Rate of P

The removal rate of P was calculated from the equation given below:

$$\text{P removal rate} = \frac{P_i - P_f}{\text{Surface area} \times \text{Treatment time}}$$

### 2.3.3. Phosphorous Uptake

The amount of P uptake was calculated from the equation given below:

$$P \text{ uptake} \left( mg \text{ container}^{-1} \right) = P \text{ conc.} \left( mg \text{ } g^{-1} \right) \times dry \text{ biomass} \left( g \text{ container}^{-1} \right)$$

### 2.3.4. Removal Efficiency of P

The removal efficiency of P was calculated from the equation given below:

$$\text{P removal efficiency } (\%) = \left[ P_i - \frac{P_f}{P_i} \right] \times 100$$

### 2.3.5. Bioconcentration Factor (BCF)

The BCF was calculated from the equation given below:

$$\text{BCF} = \text{P conc. in plant biomass} \left( \text{mg kg}^{-1} \text{ DW} \right) / \text{P conc. in water} \left( \text{mg L}^{-1} \right)$$

### 2.4. Statistical Analysis

All treatments were carried out in triplicates. Experimental data were subjected to statistical analysis using excel and SPSS. All data presented in the figures shows averages of three repetitions ± standard deviations. To see the statistical difference among the treatment, a *p*-value of 0.05 was set. A simple Pearson correlation was also applied to determine the relationship between different variables.

## 3. Results and Discussion

### 3.1. Phosphorous Removal from Solutions

The results showed that *Pistia stratiotes* and *Salvinia natans* have great potential to remove phosphorous from water solution. Figure 2A,B summarizes the phosphorous removal by *P.*

*stratiotes* from the water in both P-starved and without P-starved states. It was found that in the P-starved state, the phosphorous removal rate was high when compared to without the P-starved state. A similar pattern was observed with *S. natans* as the removal rate was higher in the starved state compared to the non-starved state (Figure 2C,D).

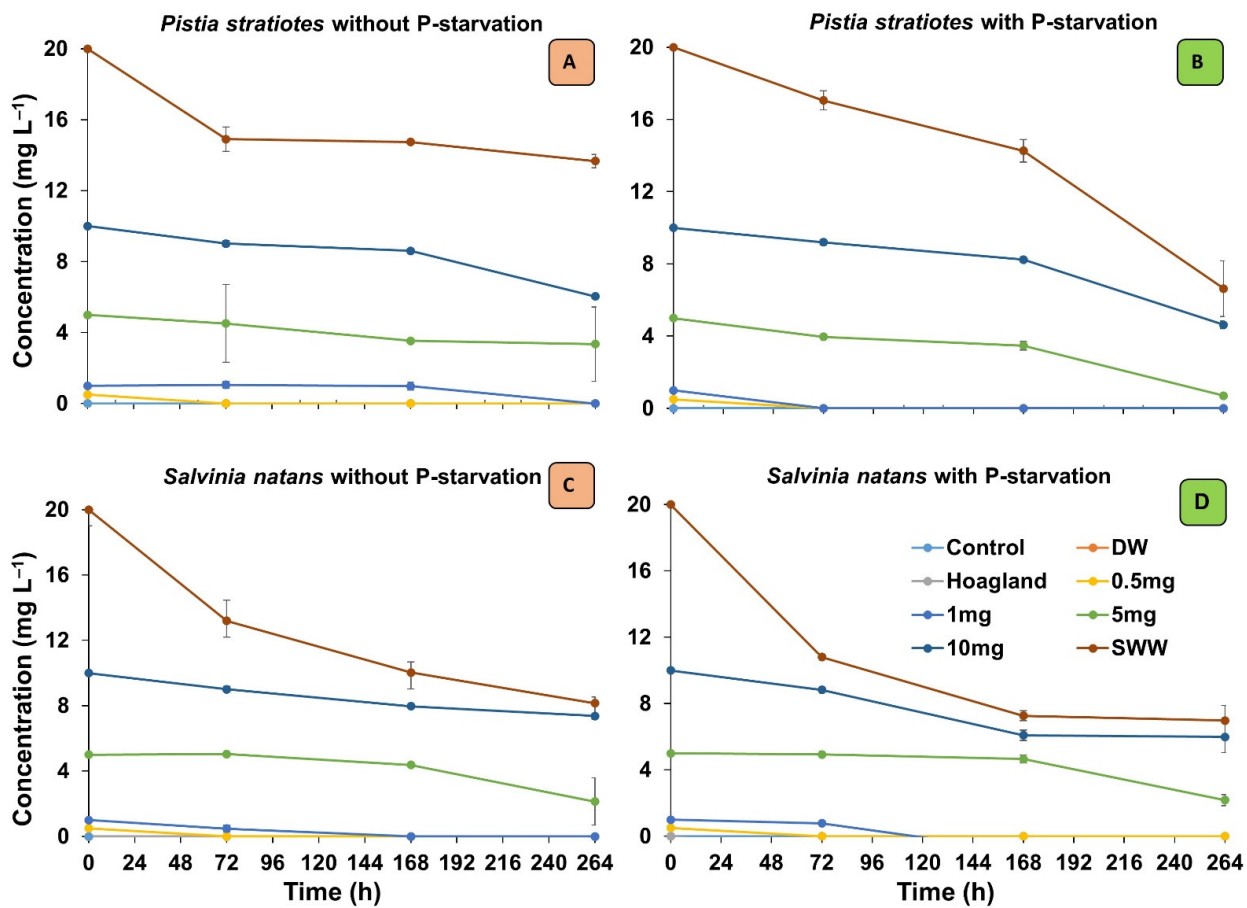

**Figure 2.** Concentration of P removed from solution by *Pistia stratiotes* without P-starvation (**A**) and with P-starvation phase (**B**), *Salvinia natans* without P-starvation (**C**) and with P-starvation (**D**); Error bars show ± SD where n = 3. SWW represents synthetic wastewater treatment, and DW represents distilled water treatment.

In low P treatment (i.e., of 0.5 mg L$^{-1}$), all P was depleted in water solution and presumably taken up by both plant species within 72 h of plant exposure. However, in 1 mg L$^{-1}$ P treatment, *Pistia stratiotes* took 168 h to remove P from the water without P-starvation. Whereas in the P-starvation treatment, *P. stratiotes* took up 72 h to remove all P from 1 mg L$^{-1}$ P treatment. Moreover, it took 264 h to remove all P from 5 mg L$^{-1}$ P treatment. By contrast, *Salvinia natans* depleted all P from 1 mg L$^{-1}$ P treatment in 72 h in both P-starved and without P-starved conditions. Without a P-starvation state, *P. stratiotes* were unable to deplete P from 5 mg L$^{-1}$ P treatment since 3.34 mg L$^{-1}$ of P was still present in the solution by end of incubation. In 10 mg L$^{-1}$ P treatment without P-starvation state treatment, *P. stratiotes* could not deplete all P since 6 mg L$^{-1}$ of P remained in the solution even after 264 h of incubation. By contrast, in the P-starved state, only 4 mg L$^{-1}$ P remained in the solution during the same incubation period. For SWW, the *P. stratiotes* removed faster P in the P-starved state since 6.6 mg L$^{-1}$ of P remained in the solution compared to 13.67 mg L$^{-1}$ of P without the P-starved phase after 264 h. A greater rate of P removal in the P-starved state may be due to the metabolic reaction that can happen in the plant body to maintain cytoplasmic phosphate, ATP, and nucleotide concentrations. Results obtained in the present study are in agreement with Shardendu et al. [24] who reported that *P. stratiotes*

significantly took up 6.12 ± 0.95 mg of phosphorous from 50 L supply. About 90% of P was removed from the medium in 60 days of incubation [24]. The same trend was observed in *S. natans*. However, P in 0.5 and 1 mg $L^{-1}$ P was depleted quickly by *S. natans* without a P-starved state for 72 h. For 5 mg $L^{-1}$ P treatment, only 2.1 $L^{-1}$ mg was left in the solution after 264 h of the incubation period. The P depletion was almost doubled in 10 mg $L^{-1}$ P and SWW treatments after 264 h of the incubation period. Only 5.99 mg of P was left in the solution in the 10 mg $L^{-1}$ P treatment and 6.79 mg of P in the SWW treatment, whereas without the P-starved state, P remained in solution was 7.37 mg in 10 mg $L^{-1}$ P treatment and 8.15 mg P SWW treatment. This result indicates that *Salvinia natans* performed better in P-starved as well as without the P-starvation phase since the SWW and 10 mg $L^{-1}$ P treatments showed a significant drop in phosphorous. Only 8 mg of P was left in the solution during 264 h in the P-starvation phase in *Salvinia natans* pot whereas there was 13 mg of P left during the same time in *Pistia stratiotes* pots containing SWW solution.

### 3.2. Phosphorous Content and Removal Rate of Plants

*P. stratiotes* accumulated 1.69, 3.04, and 5.22 mg of P from 10 mg $L^{-1}$ P treatment in the P-starvation phase after 72, 168, and 264 h, respectively. Whereas without the P-starvation phase, it could accumulate 1.21, 2.18, and 4.13 mg P after 72, 168, and 264 h, respectively (Figure 3).

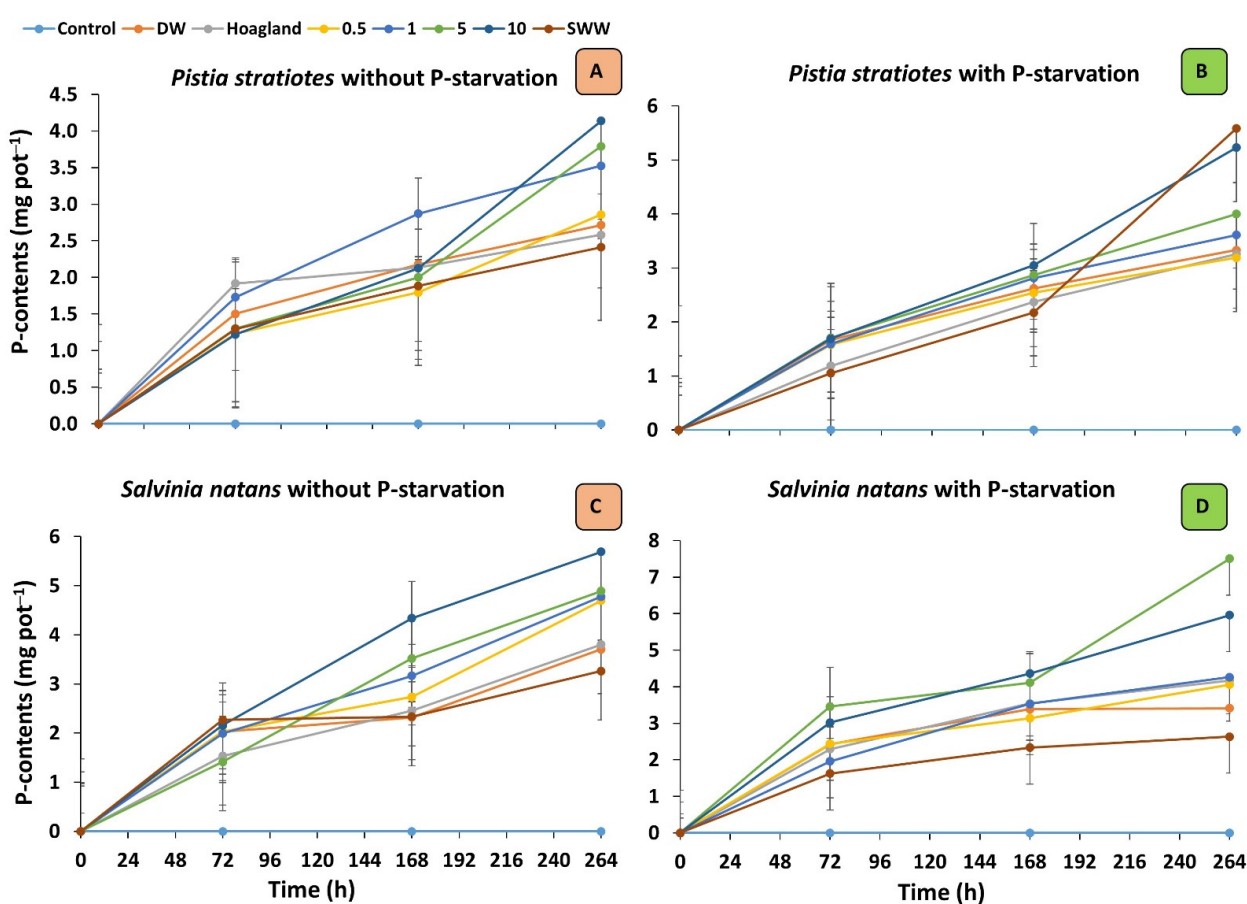

**Figure 3.** Phosphorous contents of *Pistia stratiotes* without P-starvation (**A**) and with P-starvation phase (**B**), *Salvinia natans* without P-starvation (**C**) and with P-starvation (**D**); Error bars show ± SD where n = 3. SWW represents synthetic wastewater treatment, and DW represents distilled water treatment.

There was a drastic change in SWW treatment where *P. stratiotes* accumulated 2.41 mg of P without the P-starvation phase and accumulated 5.58 mg of P in their biomass in the P-starvation phase. These results were in agreement with Kurniawan et al. [25] who

reported that *Pistia stratiotes* have the potential to remove 80 to 90% of P-content from agricultural runoff if 75% of the area that is to be treated is covered with plant for 14 days of incubation period [25]. However, this study did not test the effect of P-starvation on P accumulation. Nevertheless, *S. natans* was found to be a better accumulator compared to *Pistia stratiotes* as its P uptake was almost double in both states (with and without P-starvation). At 72, 168, and 264 h, P uptake in *S. natans* was 2, 4, and 5.6 mg, respectively, in 10 mg $L^{-1}$ treatment without P-starvation whereas, in *Pistia stratiotes*, it was 1, 2, and 4 mg, respectively. *S. natans* was able to uptake 7.5 mg P in its tissues in a P-starved state and 4.88 mg P without a P-starvation state for 264 h from 5 mg $L^{-1}$ of P treatment (Figure 3C,D). Comparison of the P-starved phase of both species in 5 mg $L^{-1}$ treatment, also showed that *S. natans* overtops the result and accumulated more P in its biomass. The removal rate of plants of both species is represented in Figure 4A–D. It was found that *S. natans* without a P-starvation state could remove 56.6% of P in 36 h of the incubation period. *P. stratiotes* and *S. natans* showed a similar trend in P-removal rates with and without P-starved phases.

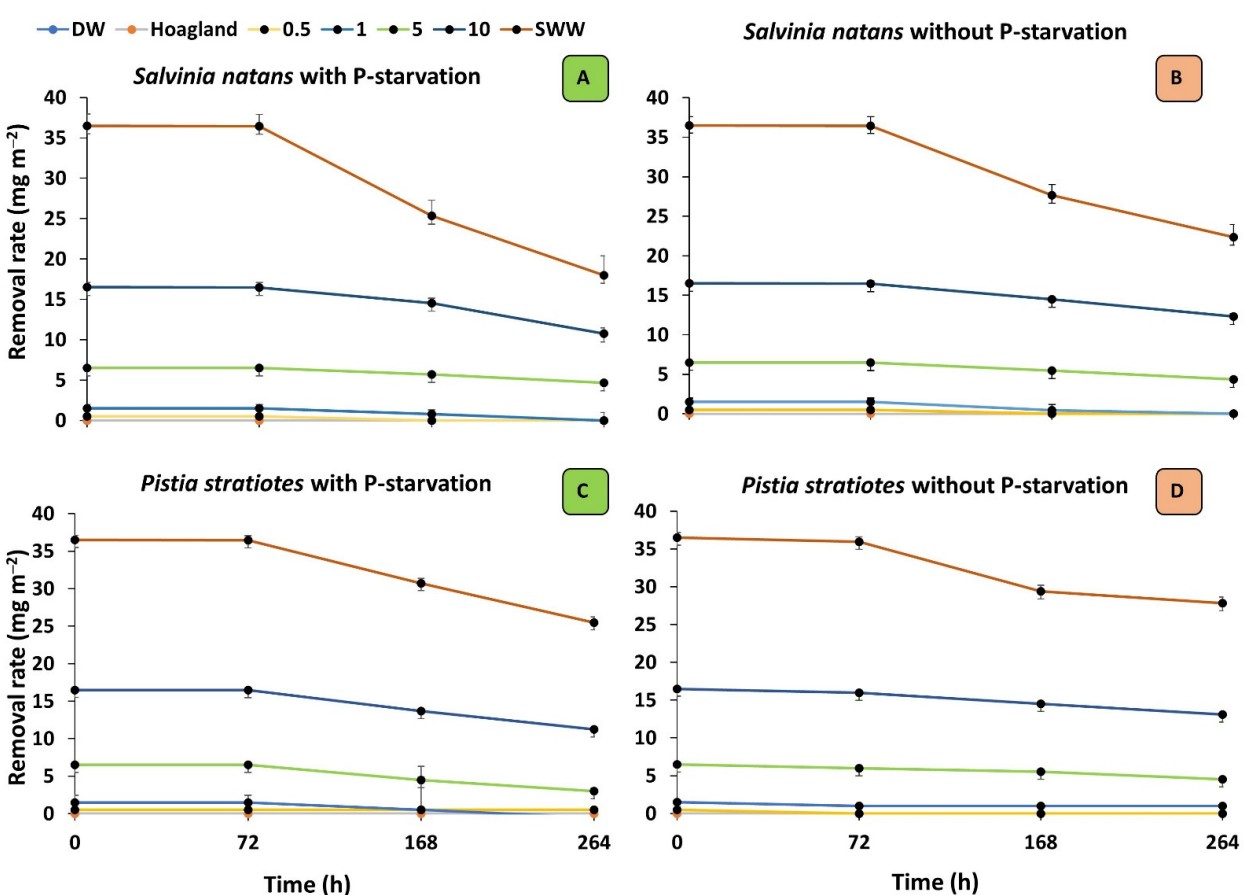

**Figure 4.** Removal rate of *Pistia stratiotes* during with P-starvation (**A**) and without P-starvation phase (**B**), *Salvinia natans* with P-starvation (**C**) and without P-starvation (**D**); Error bars show ± SD where n = 3. SWW represents synthetic wastewater treatment, and DW represents distilled water treatment.

### 3.3. Removal Efficiency of Plants

*Salvinia natans* showed the highest phosphorous removal efficiency compared to *P. stratiotes* due to its small and mat-like growing pattern, which occupied more space in the surface water. Both species have a great affinity for phosphorous. Both species completely (100%) depleted the 0.5 and 1 mg P within 72 h. This might be because phosphorus is the second essential macronutrient after N required by plants for their optimum growth and development. Phosphorus removal efficiency by both aquatic plants with and without P-starvation is represented in Figure 5A–D.

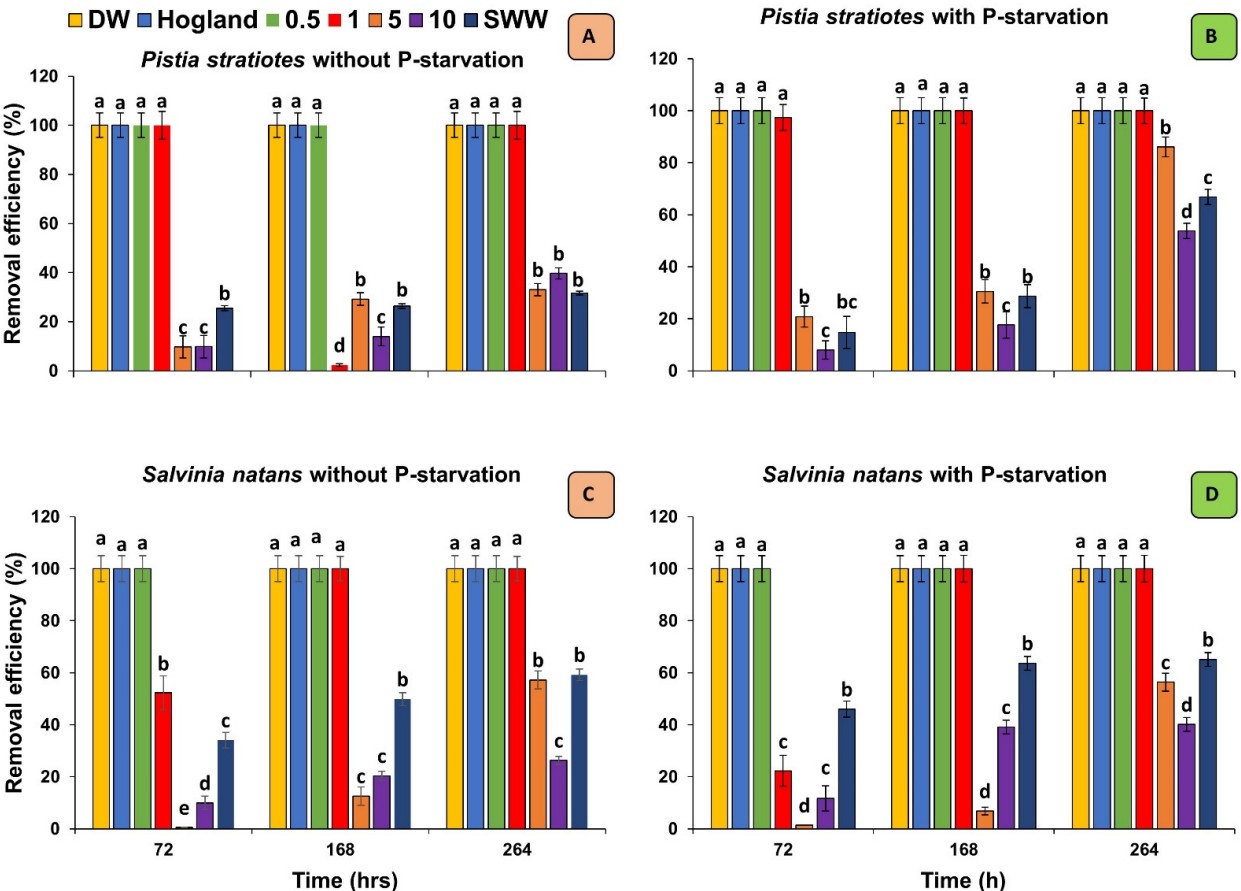

**Figure 5.** Removal efficiency of *Pistia stratiotes* without P-starvation (**A**) and P-starvation phase (**B**), *Salvinia natans* without P-starvation (**C**) and with P-starvation (**D**); Error bars show $\pm$ SD where n = 3. Under each incubation time, the bars with different letters are significantly different from each other at $\alpha = 0.05$. SWW represents synthetic wastewater treatment, and DW represents distilled water treatment.

The *P. stratiotes* removed 86.04% of P from 5 mg $L^{-1}$ P solution, 53.76% from 10 mg $L^{-1}$ P solution and 66.84% from SWW in the P-starved phase whereas, removal efficiency without the P-starvation phase was 33.03% from 5 mg $L^{-1}$ P solution, 39.66% from 10 mg $L^{-1}$ P solution, and 31.64% from SWW after 264 h interval. Compared to *S. natans*, it removed 86% P from a 5 mg $L^{-1}$ P solution, whereas *S. natans* removed 56% when exposed to the same P solution. In another study, it was found that *P. stratiotes* were capable of removing 81.6% of P from sewage waters during 10 day incubation period [26]. The *S. natans* removed 40.09% of P from 10 mg $L^{-1}$ P solution and 65.12% from SWW in a P-starved state, whereas without the P-starvation state, the values obtained were 26.28% from 10 mg $L^{-1}$ P treatment and 59.22% from SWW treatment. *Salvinia natans* performed well in SWW solutions compared to *Pistia stratiotes*. *Salvinia natans* removed 45% P at 72 h, 63.6% P at 168 h, and 65% P at 264 h whereas *Pistia stratiotes* removed 14.7% P at 72 h, 28% P at 168 h, and 66.8% P at 264 h. In another study, Salvinia showed the highest phosphorous removal efficiency (14.2%) in artificial wastewater when compared to different aquatic plant species like *Ipomoea aquatica*, *Eleocharis plantagineiformis*, *Colocasia tonoimo*, *Dysophylla sampsonii*, *Typha orientalis*, *Hydrocotyle vulgaris* and *Eleocharis plantagineiformis* [21]. Qin et al. [27] evaluated *Eichornia crassipes* and *Pistia stratiotes* L. for phytoremediation of wastewater and reported that *Pistia stratiotes* L. showed higher P removal efficiency that can be attributed to better acquisition and uptake of P due to longer roots. Similarly, Yu et al. [28] investigated six macrophytes and reported that *I. sanguinea* was more efficient in treating eutrophic water with N and P. Nevertheless, these authors did not investigate the potential of these aquatic species in the P-starved conditions which we tested in this study [28].

### 3.4. Bioconcentration Factor (BCF)

The bioconcentration factor is calculated to find out the ability of the macrophytes to remove contaminates (metal, minerals, nutrients, etc.) from the substrate (soil, water, air). Plants having BCF value > 1 can be considered hyperaccumulators [29].

The results (Figure 6A–D) indicated that both plants may not be used as hyperaccumulators from solutions with high P concentration because the values obtained were less than 1, i.e., in the range of 0.1–0.8 for 5 mg $L^{-1}$, 10 mg $L^{-1}$, and SWW (20 mg $L^{-1}$) of P-containing solutions. Whereas for low concentrations, i.e., 0.5 mg $L^{-1}$ and 1 mg $L^{-1}$ and in 5 mg $L^{-1}$ (in the case of *Salvinia natans*) the BCF values were greater than 1 and can be classified as hyperaccumulators. BCF values of *Salvinia natans* were greater than those of *Pistia stratiotes*. The *P. stratiotes* concentrated (9.07) maximum of P from 0.5 mg $L^{-1}$ P solution, (5.68) from 1 mg $L^{-1}$ P, (0.94) from 5 mg $L^{-1}$ P, (0.614) from 10 mg $L^{-1}$ P, and (0.31) from SWW in without P-starved phase after 264 h of exposure, whereas in the P-starvation phase the values were (9.97) from 0.5 mg $L^{-1}$ P, (6.05) from 1 mg $L^{-1}$ P, (1.34) from 5 mg $L^{-1}$ P, (0.79) from 10 mg $L^{-1}$ P, and (0.56) from SWW. By contrast, *S. natans* concentrated (9.6) P from 0.5 mg $L^{-1}$ P, (5.21) from 1 mg $L^{-1}$ P, (0.71) from 5 mg $L^{-1}$ P, (0.616) from 10 mg $L^{-1}$ P and (0.47) from SWW in without P-starved phase at 264 h. In the P-starvation phase, the values were (10.5) from 0.5 mg $L^{-1}$ P, (5.89) from 1 mg $L^{-1}$ P, (1.47) from 5 mg $L^{-1}$ P, (0.81) from 10 mg $L^{-1}$ P and (0.42) from SWW. Both species performed better in the P-starved conditions whereas *Salvinia natans* concentrated more P into their biomass. Results align with earlier studies that also showed that *Salvinia natans* and *Pistia stratiotes* are excellent hyperaccumulators for heavy metals and minerals [30–32].

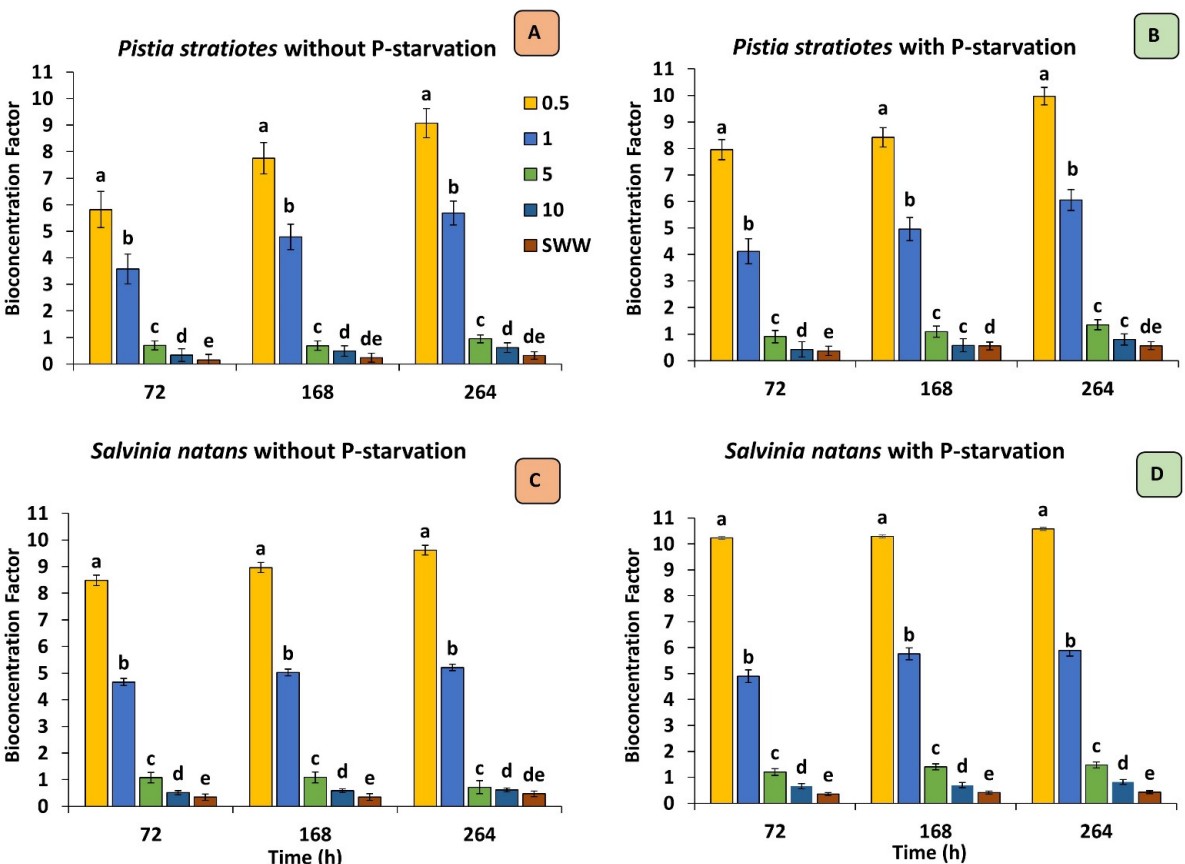

**Figure 6.** Bioconcentration factor of *Pistia stratiotes* during without P-starvation (**A**) and with P-starvation phase (**B**), *Salvinia natans* without P-starvation (**C**) and with P-starvation (**D**); Error bars show ± SD where n = 3. Under each incubation time, the bars with different letters are significantly different from each other at α = 0.05. SWW represents synthetic wastewater treatment, and DW represents distilled water treatment.

Statistically significant correlation coefficients between P removal amount and P-concentrations in both plant species with 5 and 10 mg $L^{-1}$ P solutions are shown in Figures 7 and 8.

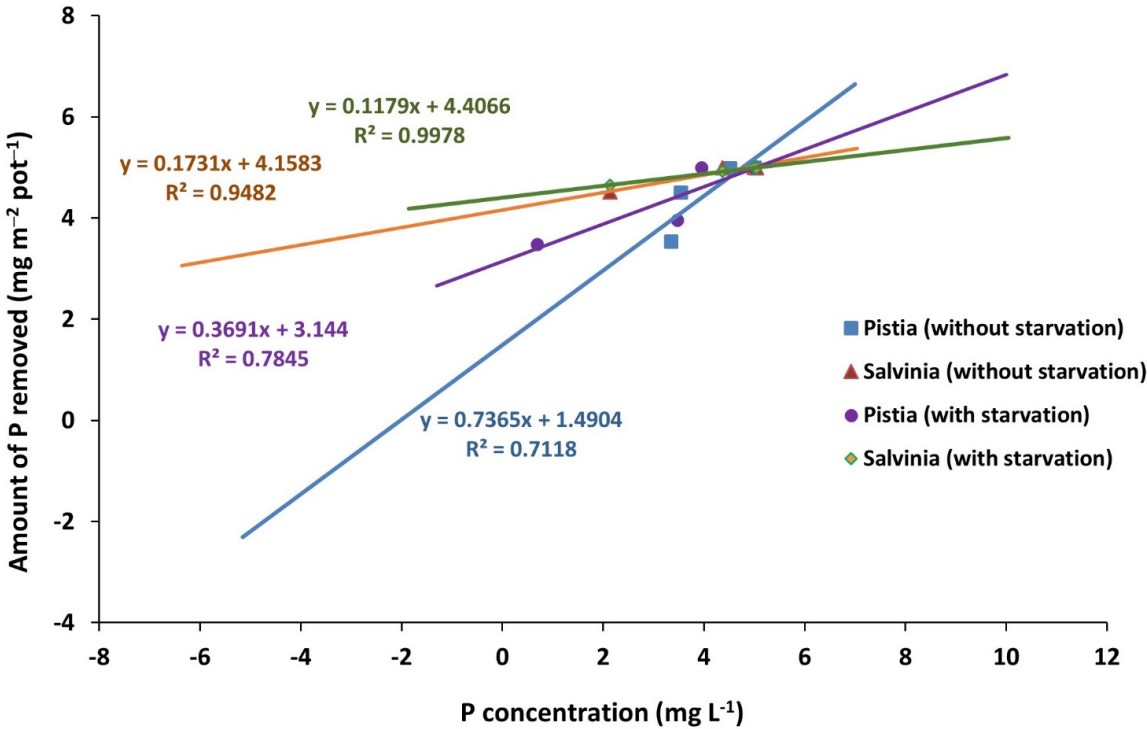

**Figure 7.** Relationship between amount of P-removed (mg $m^{-2}$ $pot^{-1}$) and P concentration in plants (mg $L^{-1}$) at 5 mg $L^{-1}$.

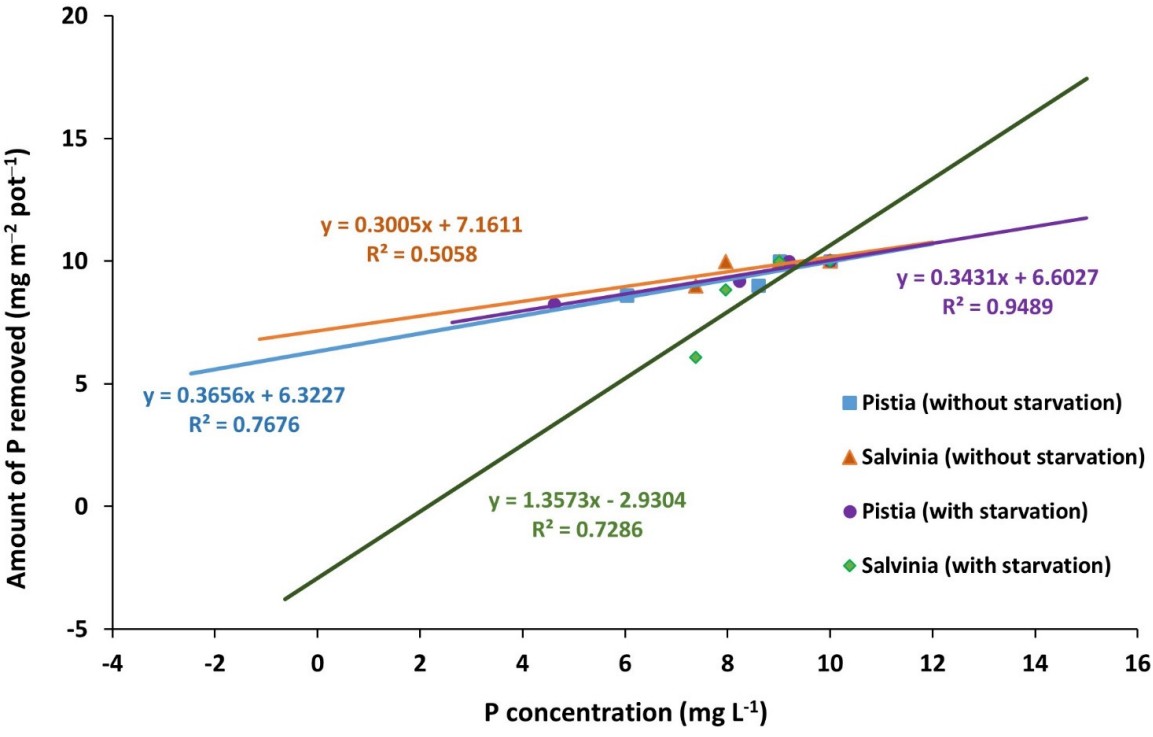

**Figure 8.** Relationship between amount of P-removed (mg $m^{-2}$ $pot^{-1}$) and P-concentration in plants (mg $L^{-1}$) at 10 mg $L^{-1}$.

At 5 mg L$^{-1}$ P-induced eutrophic water, the *Salvinia natans* were found to be more efficient as compared to *Pistia stratiotes* in accumulating P in their biomass. When exposed to 10 mg L$^{-1}$, the *Pistia stratiotes* accumulated slightly more P compared to *Salvinia natans*. All correlations were close to 1 which signifies that P removed from solutions was accumulated into plants biomass. This correlation indicated that P-removed from P-eutrophicated solutions was accumulated in plant biomass. This type of phytoremediation is termed phytoextraction or phytoaccumulation, where contaminants, predominantly non-degradable inorganics, are removed by the plants. Therefore, based on the results obtained in the present experiments, it is plausible to conclude that *Salvinia natans* had a higher phytoremediation ability as compared to *Pistia stratiotes* to remove P from P-eutrophicated water and can decrease P-concentration significantly in aquatic ecosystems. Su et al. [21] also reported that *S. natans* showed the highest total phosphorous removal efficiency (67.3%) in highly polluted wastewater when compared with other tested species. In another study, *S. natans* removed 22.4 ± 0.1 P from non-aerated and 54.4 ± 0.1 P from aerated sewage water collected from Bhagwanpur, Varanasi, India [30]. Reddy & de Busk [18] exposed *S. natans* to stimulated synthetic wastewater and observed that *Salvinia natans* was able to remove 9.0 ± 2.2 P from June to September and 10.7 ± 1.1 P from December to February. Al-Hamdani et al. [33] reported that the highest uptake of P into the salvinia tissues resulted when phosphorous levels were elevated 100-fold (10 mg L$^{-1}$) and the highest yield was obtained after 14 days of exposure. For the genus Salvinia, phosphorus values around 100 mg P/m$^2$/d for absorption rates and a percentage of reduction between 35% and 50% were reported [34]. In semi-arid climates like that of Algeria, Ayache et al. [35] also reported that *S. natans* removed 37% of P from water solutions.

### 3.5. Biomass of Plants

In general, the productivity of macrophytes is higher than that of terrestrial plants. Macrophytes have a high tolerance for fluctuations in environmental conditions and show high photosynthetic efficiencies. The uptake of nutrients by macrophytes is essential for their growth and reproduction. The high productivity of macrophytes enables substantial amounts of nutrients to be stored in plant biomass. In the present experiments, measurements of biomass were made after 72, 168, and 264 h without P-starved and P-starved phases. In the without P-starvation phase, the *P. stratiotes* produced average biomass of 18 g in 264 h of incubation, and in the P-starvation phase, it produced average biomass of 14 g due to the P stress environment. However, the uptake of P was more in P-starved plants compared to non-P-starved plants. Nevertheless, *S. natans* showed a similar trend due to P stress. However, when it was exposed to 10 mg L$^{-1}$ of P for 264 h of incubation, 21 g of fresh biomass was recorded during the P-starved phase, more than *P. stratiotes* under similar conditions. In another study, it was found that *S. natans* resulted in the highest relative and absolute fresh biomass [21]. It covers the entire area with its mat-like characteristics and absorbs phosphorous.

To starve the plants, they were placed in distilled water for 3 days. This stimulated the stress in some plants and they started shedding roots, especially since it was highly visible in *P. stratiotes* due to its long roots. However, when they were placed in P-fed waters, the roots started to grow again. This indicated the positive response of the plants to phosphorous-induced waters and their ability to gain reasonable biomass within a short period. This abrupt biomass gain was more prominent in *S. natans* in the P-starved state. Both species showed a gradual increase in phosphorous and enhanced their biomass. Biomass was increased by three folds during 264 h of incubation time. Due to more contact time with the P-enriched water, the plants were able to remove maximum phosphorous from the aquatic medium. However, the negative effect was seen with the plants grown in synthetic wastewater treatment (SWW). Both species started to die after 72 h of the time interval. The *P. stratiotes* shed its roots and after 168 h both species started to dissolve in the SWW solution. This might be due to the stress of heavy metals or salts present in SWW. In another study, when *P. stratiotes* were exposed to rich cadmium-rich waters for 21 days, the

plants failed to grow and showed signs of toxicity like necrosis when placed in a 15 mg L$^{-1}$ cadmium-treated solution [36]. However, despite the stress, the plants still managed to uptake phosphorous. Furthermore, the growth of the plants can be restricted due to spatial constraints, as was observed while culturing these fast-growing plants. The plants were cultured in 6 tubs initially in December but at the end of June 2021, 20 tubs were covered by these plants. Spatial constraints can initiate necrosis in plants as well.

*Pistia stratiotes*, being a free-floater needs to be harvested from time to time. Once water P levels are reduced to normal, the plant body may be harvested to use as animal feed or as biofuel and healthy biodiversity may be allowed to develop in the restored wetland [18]. When the *S. natans* coverage surpasses 85%, the *S. natans* population can significantly reduce the oxygen levels of the water body and leads to an anaerobic state.

## 4. Conclusions

The salvinia floater and water lettuce both showed differential growth response and the ability to uptake phosphorous when exposed to different concentrations and different incubation times. The salvinia floater had the highest phosphorous-removing ability, i.e., 86.0% P from a 5 mg L$^{-1}$ P solution due to its mat-like structure, while *S. natans* removed only 56.6% P when exposed to the same P concentration solution. Moreover, both plants with P-starvation showed high phosphorous removal compared to plants without P-starvation. Thus, for effective P-removal, P-starved plants can be deployed effectively for efficient removal of P from aquatic environments. This study proved that aquatic plants are a possible and low-budget treatment for P-polluted water bodies. For phytoremediation on large scale, subsidies should be provided to farmers, limnologists, and environmentalists to introduce constructed wetlands near farms. After treatment, the plants can be used as a nutrient source (compost) or soil conditioner (biochar), livestock feed, or biofuel. This study, however, was conducted on a laboratory scale using microcosms, therefore, these plants should be tested on the field scale for a better understanding of their potential to remove P from water solutions.

**Author Contributions:** Conceptualization, S.D. and M.S.A.; Data curation, S.D., S.A. and A.D.; Formal analysis, S.D., M.V., S.A. and A.D.; Funding acquisition, M.V. and A.D.; Investigation, S.D.; Methodology, M.S.A.; Project administration, M.S.A., M.V. and A.D.; Resources, S.D., M.S.A., M.V., S.A. and A.D.; Software, S.D., M.S.A., M.V., S.A. and A.D.; Supervision, M.S.A.; Validation, S.D., M.S.A., M.V., S.A. and A.D.; Visualization, S.D., M.S.A., M.V., S.A. and A.D.; Writing—original draft, S.D.; Writing—review and editing, M.S.A., M.V., S.A. and A.D. All authors have read and agreed to the published version of the manuscript.

**Funding:** This research received no external funding.

**Institutional Review Board Statement:** Not applicable.

**Informed Consent Statement:** Not applicable.

**Data Availability Statement:** Not applicable.

**Acknowledgments:** The authors greatly acknowledge the technical support of Derk Bakker for installing artificial lights and Rafaqat Mirza for his support in lab work.

**Conflicts of Interest:** The authors declare no conflict of interest.

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
