# Peer review of "Microcosm Study on the Potential of Aquatic Macrophytes for Phytoremediation of Phosphorus-Induced Eutrophication"

_sustainability, doi:10.3390/su142416415_

Round 1

Reviewer 1 Report

The manuscript well describes the research of the effectiveness of two aquatic macrophytes species and water lettuce for phosphorous removal through serial microcosm experiments. This is not a groundbreaking research, although it may bring interesting conclusions. There are no specific conclusions. This chapter is a casual discussion. The abstract is badly written. Provides background information on the subject and does not summarize the content of the manuscript.

Author Response

Response to Reviewer 1 comments

Comment: The manuscript well describes the research of the effectiveness of two aquatic macrophytes species and water lettuce for phosphorous removal through serial microcosm experiments. This is not a groundbreaking research, although it may bring interesting conclusions. There are no specific conclusions. This chapter is a casual discussion. The abstract is badly written. Provides background information on the subject and does not summarize the content of the manuscript.

Response: All the concerns raised by the reviewer have been incorporated in the revised manuscript. Abstract, Results and Discussion, Figures, and other sections have been thoroughly revised. The quality of all Figures has been improved as per the suggestions of the reviewer.

Author Response

Response to Reviewer 2 comments

Comment: Dear Editor, Thank you for providing me the chance to review the manuscript titled “Microcosm study on the potential of aquatic macrophytes for phytoremediation of phosphorus induced eutrophication”. The work presents the information related to phosphorus removal by two macrophytes Salvinia natans and Pistia stratiotes. The present condition of manuscript is very weak, especially method, and results and discussion need substantial improvements. Therefore, I do not recommend this paper to be considered for sustainability. The authors must improve the quality of manuscript, and for this reason I am also sharing a few comments.

Response: All the concerns raised by the reviewer have been incorporated in the revised manuscript. Abstract, Results and Discussion, Figures, and other sections have been thoroughly revised. The quality of all Figures has been improved as per the suggestions of the reviewer.

Comment: Line 14-15. Distribute it in 2 separate statements

Response: This correction has been incorporated.

Comment: Line 17: replace with “filter P from water contaminated with high levels of nutrients.”

Response: Done

Comment: Line 19 and 20: replace“P-removal rates and efficiency”

Response: Done

Comment: Line 20-21: I think it’ s unnecessary to mention “exposed to modified Hoagland solution”

Response: Agreed and modified as needed.

Comment: Line 22-24: this sentence was already mentioned earlier, consider removing this reptation

Response: All repetitions as mentioned by the reviewer have been removed.

Comment: Line 14-33: the abstract need reformulation with authors presenting their most significant results. At present it is very general and nothing very interesting is available.

Response: Abstract has been revised as per the comments of the reviewers.

Comment: Line 53: please provide relevant reference for the statement “Phosphorus is a nonrenewable resource that is expected to deplete in 100 to 300 years if not used efficiently.”

Response: Reference has been provided to support this statement.

Comment: Line 57 to 59: consider revising this. What do authors mean by water like wetlands? It really does not give any correct indication what authors wanted to establish here.

Response: This sentence has been restructured to avoid confusion.

Comment: Line 65: authors have already introduced (P) earlier, hence no need to reintroduce this.

Response: This repetition has been removed.

Comment: Line 63-72: authors have mostly repeated the commentary of first and second paragraph of the same manuscript, consider either removing or replacing them in the 1st or 2nd paragraph.

Response: All repetitions in this section as mentioned by the reviewer have been removed.

Comment: Authors should mention a few methods that are used for P removal, and also establish the reason why using plant is comparatively a good option than the rest.

Response: This is mentioned in our manuscript that the use of plants such as macrophytes is an environmentally and economically viable option for P removal.

Comment: Line 74-86: authors are repeating the same material again. Consider revising this whole paragraph

Response: All repetitions as mentioned by the reviewer have been removed.

Comment: Line 104: Author probably have to reformulate this statement “Pistia stratiotes might be an efficient macrophyte for accumulating P because it can double biomass within a month.”

Response: This sentence has been corrected.

Comment: There are numerous studies that used the same plant and hance it can be inferred that for the removal of nutrient like N and P Pistia stratiotes have been used extensively. This brings to another issue. Author should also provide the novelty statement very clearly for this research work.

Response: We tested these plants in P -starved and P-enriched conditions which is the novelty of our study.

Comment: Line 116: what do authors mean by “10, 13, and 17 plants were used” did they mean plants per tub? Or per treatment.

Response: This sentence has been elaborated to avoid confusion.

Comment: Line 149: what is DIP

Response: This sentence is removed as was unnecessary.

Comment: Line 163: three replications of plants or three replications of treatment?

Response: This is corrected since three replications of treatments.

Comment: Line 167: Author must mention why they used synthetic wastewater. Further they must also mention how the said concentration of salts were being choose. The composition seems incomplete as in table 1 there is no indication if this was per liter or per gallon.

Response: Synthetic wastewater was used for this study to investigate the response of tested plants exposed to P in presence of other metals. Salt concentrations in synthetic wastewater were used by considering their toxic doses in conventional wastewater sources. Tables 1 and 2 have been modified and the correct unit depicting concentrations of different salts is added.

Comment: Line 170: The composition of Hoagland solution seems wrong; authors are advised to recheck this composition.

Response: Modified Hoagland solution used in this study was 1% of the original composition.

Comment: Line 193: is there any reason why authors choose to harvest plant exactly at 3 pm? If there is no explanation, authors should avoid the unnecessary word jargon.

Response: Agreed and this has been corrected.

Comment: Line 195: just 3, 7, and 11 days were enough to see the drastic changes in biomass?

Response: These are sufficient time durations to see a change in biomass based on our previous studies [1].

Comment: Line 207-211: authors must provide the reference to this method adopted

Response: Reference has been added.

Comment: Line 213: the detail for operational condition and equipment used for the spectrophotometry is missing

Response: Proper citation to support this comment is added in the manuscript.

Comment: Line 249: authors must mention the information about mesocosm. As if the volume is increasing this removal efficiency will not be achievable.

Response: Mesocosms used in this study were 1L, and we maintained this volume throughout the incubation.

Comment: Line 256: the quality of figure is very poor. Please avoid repetition of the same figure legend in all the sub figure. Further use a same template for all sub figure, the figure 2.B seems to have extra grey boundary while other do not have.

Response: All figures have been revised incorporating all points mentioned in this comment. Extra legend and the borders have also been removed.

Comment: Line 248 to 460: while results and discussion section is very weak. All figures need revision, as the graphs presented aren’ t following the same pattern. There is some serious lack of results discussion, which make it difficult to take any idea what the possible explanation of the results could be or how they are linked with other findings. Also, add the limitation of the study, along with the future perspective of this work in the last paragraph of results and discussion.

Response: Results and discussion section has been revised as per the comment of the reviewer. All figures have been improved.

Reviewer 3 Report

Overall content is okay. But objectives and neccessity of the work is missing in abstract, inttoduction and conclusion.

Please revise your manuscript grammatically.

Author Response

Response to Reviewer 3 comments

Comment: Overall content is okay. But objectives and neccessity of the work is missing in abstract, introduction and conclusion.

Response: The manuscript has been revised thoroughly as per the instructions of the reviewer.

Comment: Please revise your manuscript grammatically.

Response: The manuscript was proofread again to remove grammatical mistakes.

Round 2

Reviewer 1 Report

Accept

Author Response

Thanks for your recommendation

Author Response

Response to Reviewer 2 comments

Dear Editor,

The manuscript title “Microcosm study on the potential of aquatic macrophytes for phytoremediation of phosphorus-induced eutrophication”, have been improved significantly. The present condition of manuscript is much better than the previous submission. I recommend this paper to be considered for sustainability, however before this the authors still must do some minor changes, that they can find below.

Response: Thanks for your recommendations. We have incorporated the suggested changes

Comment: Line 20: mention the incubation time in brackets.

Response: We have mentioned the incubation time in brackets

Comment: Line 21-23: The statement is too general. Please mention some numeric value indicating the amount of biomass produced by the respective plant, and mention the removal percentage

Response: We have added the numeric values indicating the amount of biomass produced by the respective plant, and mentioned the removal percentage as per the suggestion of the reviewer

Comment: Line 53: I suggest that authors should replace p-containing wetland, with excessive P contaminated wetlands. Reason is at least P is an integral part of wetland soil and sentiments.

Response: We have replaced “P-containing wetlands” with “excessive P-contaminated wetlands” as per the suggestion of the reviewer

Comment: Line 70 to 81: In the revised submission authors have removed the paragraph altogether, however in the previous review authors were advised to mention a few methods that are used for P removal and establish the reason why using plant is comparatively a good option than the rest. This suggestion has been ignored by the author. This must be incorporated in the present line 70 to 81 paragraph

Response: We apologize for this ignorance. We have now added the reasons why using the plant is comparatively a good option than the rest

Comment: Line 149: The source and the activities that results in the production of wastewater play a significant role in the composition of wastewater. Though the provided information is improving the uncertainty, which was there in the last submission, however, at present authors must mention the sources analyzed and provide this information in the supplementary information.

Response: The synthetic wastewater was prepared and used after considering the average metal and pollutant concentration in domestic wastewater collected and tested from the surrounding drains and sewerage system of Lahore, Pakistan (Hamid et al. 2013).

Hamid, A., Zeb, M., Mehmood, A., Akhtar, S., Saif, S., 2013. Assessment of Wastewater Quality of Drains for Irrigation. Journal of Environmental Protection, 4 (9): 937-945. doi: 10.4236/jep.2013.49108.

Comment: Line 241: The quality of figures has been sufficiently improved. Just one minor correction, please use the same color scheme for the same treatments as the scheme used in figure 1 is different than that of 2 and 3. Also the legend of figure 1 should be revised as of figure 2 and 3. In figure 2 some points are missing the positive caps of SD. Further the x axis should be similar in all figures, as the reading were taken at the same interval (isn’t it?). I suggest only mention the point at which the values were taken as authors themselves done in figure 3.

Response: We have revised the Figures by the same color for the same treatments in all three figures and made the x-axis at similar intervals

Comment: Line 452: please mention the percentage for “The salvinia floater has the highest phosphorous-removing ability, that was …% higher than?”

Response: We have revised the sentence as “The salvinia floater had the highest phosphorous-removing ability i.e., 86.0% P from a 5 mg L-1 P solution due to its mat-like structure, while S. natans removed only 56.6% P when exposed to the same P concentration solution.” As per the suggestion of the reviewer